# Coherent X-rays reveal the influence of cage effects on ultrafast water dynamics

Fivos Perakis [1,2], Gaia Camisasca [1], Thomas J. Lane[2], Alexander Späh[1], Kjartan Thor Wikfeldt[1],
Jonas A. Sellberg [3], Felix Lehmkühler [4,5], Harshad Pathak [1], Kyung Hwan Kim[1], Katrin Amann-Winkel [1],
Simon Schreck[1], Sanghoon Song [2], Takahiro Sato[2], Marcin Sikorski[2,6], Andre Eilert [2], Trevor McQueen[2],
Hirohito Ogasawara [2], Dennis Nordlund [2], Wojciech Roseker[4], Jake Koralek[2], Silke Nelson[2], Philip Hart[2],
Roberto Alonso-Mori [2], Yiping Feng[2], Diling Zhu[2], Aymeric Robert [2], Gerhard Grübel[4,5],
Lars G.M. Pettersson [1] & Anders Nilsson[1]

The dynamics of liquid water feature a variety of time scales, ranging from extremely fast ballistic-like thermal motion, to slower molecular diffusion and hydrogen-bond rearrangements. Here, we utilize coherent X-ray pulses to investigate the sub-100 fs equilibrium dynamics of water from ambient conditions down to supercooled temperatures. This novel approach utilizes the inherent capability of X-ray speckle visibility spectroscopy to measure equilibrium intermolecular dynamics with lengthscale selectivity, by measuring oxygen motion in momentum space. The observed decay of the speckle contrast at the first diffraction peak, which reflects tetrahedral coordination, is attributed to motion on a molecular scale within the first 120 fs. Through comparison with molecular dynamics simulations, we conclude that the slowing down upon cooling from 328 K down to 253 K is not due to simple thermal ballistic-like motion, but that cage effects play an important role even on timescales over 25 fs due to hydrogen-bonding.

[1] Department of Physics, AlbaNova University Center, Stockholm University, S-106 91 Stockholm, Sweden. [2] SLAC National Accelerator Laboratory, 2575 Sand Hill Road, Menlo Park, California, CA 94025, USA. [3] Biomedical and X-ray Physics, Department of Applied Physics, AlbaNova University Center, KTH Royal Institute of Technology, S-10691 Stockholm, Sweden. [4] Deutsches Elektronen-Synchrotron DESY, Notkestr. 85, 22607 Hamburg, Germany. [5] Hamburg Centre for Ultrafast Imaging, Luruper Chaussee 149, 22761 Hamburg, Germany. [6] European XFEL, Holzkoppel 4, 22869 Schenefeld, Germany. Correspondence and requests for materials should be addressed to F.P. (email: f.perakis@fysik.su.se) or to A.N. (email: andersn@fysik.su.se)

Water is a complex and anomalous liquid, despite the simple structure of the individual water molecule. This is due to the ability of water molecules to form hydrogen bonds (H-bonds), giving rise to a highly disordered three-dimensional network, which in turn leads to peculiar thermodynamic and structural properties[1,2]. The H-bond network fluctuates on multiple lengthscales, resulting in rich and heterogeneous dynamics, which are yet to be fully understood. The translational and rotational diffusion of water can be accessed through quasi-elastic neutron scattering[3–5] and neutron spin-echo[6,7] measurements by deconvolving the hydrogen and oxygen contributions to the dynamics. In addition, the diffusion dynamics of liquid water can be probed by inelastic X-ray scattering experiments in the frequency domain[8]. Based on the correlation between the OH stretch frequency fluctuation and changes in the local H-bond environment in ultrafast vibrational spectroscopic investigations[9–13] the H-bond breaking and forming dynamics have been proposed to occur on a picosecond timescale. Furthermore, the intermolecular dynamics of water molecules can be probed in the THz regime, where the low-frequency modes in the range 50–300 $cm^{-1}$ are attributed to H-bond oscillations[14]. The sensitivity of these THz modes to the local H-bond network has been seen both from simulations[15] and experiments[16,17]. One of the challenges when using spectroscopic techniques, both in the IR- and THz-regime, is relating the spectroscopic observable to a specific lengthscale and motion in the liquid. The temporal resolution of most pump-probe implementations is also limited due to the longer wavelengths in the IR and THz-regime as compared to X-rays, which makes it difficult to probe the sub-100 fs regime where initial molecular displacements occur.

Here, we demonstrate an alternative approach to probe equilibrium dynamics, utilizing a time-domain approach which at the same time provides momentum-space resolution by coherent X-ray scattering. By extending X-ray Photon Correlation Spectroscopy (XPCS)[18–22] in the ultrafast regime, we utilize the unique temporal resolution of X-ray Free-Electron Laser (FEL) sources to probe the dynamics of water molecules in the sub-100 fs regime. This approach is often referred to as X-ray Speckle Visibility Spectroscopy (XSVS)[23–25], which enables the measurement of dynamics within the exposure time of the experiment. This can be achieved due to the nearly full transverse coherence of the FELs, which allows resolving single-shot X-ray speckle patterns[26]. In our experiment, we tune the exposure time from 10 to 120 fs by adjusting the X-ray pulse duration and observe a reduction of speckle contrast attributed to molecular motion. By measuring the dynamics at wide-angle scattering wave vectors ($Q = 1.95\,Å^{-1}$) we probe motion on atomic lengthscales, which at the first diffraction peak of water relates to changes in local tetrahedral coordination[27]. We compare the experimental results to those obtained from molecular dynamics (MD) simulations (TIP4P/2005[28] and MB-pol[29]) as well as with a ballistic model depicting thermal motion. We conclude that the observed dynamics deviate from a purely ballistic regime already after 25 fs, which is well reproduced by MD simulations. From the observed strong temperature dependence, we argue that the slowing down of motion upon cooling is affected by caging effects, which can be related to the increased tetrahedral coordination in the super-cooled regime.

## Results

**Ultrafast XSVS**. To perform XSVS we utilize the X-ray Correlation Spectroscopy (XCS) instrument[30] at the Linac Coherent Light Source (LCLS). A schematic of the experiment is shown in Fig. 1a. Ultrashort X-ray pulses are used with variable pulse duration $\delta t$, ranging from 10 to 120 fs. Water droplets are injected into the experimental chamber and their temperature is varied over the range $T = 253–328\,K$ (see Methods section)[27,31]. A Cornell-SLAC Pixel Array Detector[32] (CSPAD) is used to record X-ray diffraction patterns over a broad momentum transfer range

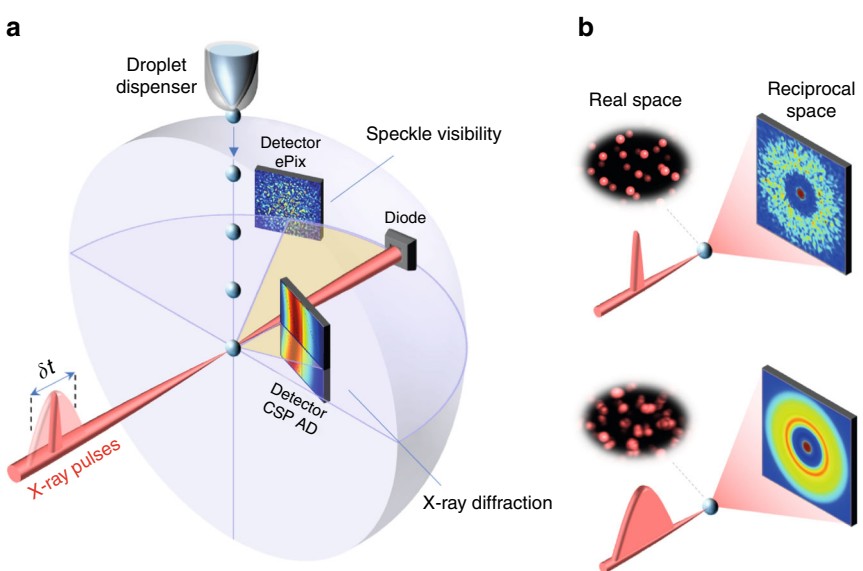

**Fig. 1** Coherent X-ray diffraction of water droplets with variable pulse duration. **a** Schematic of the experimental setup used at LCLS to measure liquid droplets utilizing two detectors at different sample-detector distances. A CSPAD detector in close proximity to the sample is used to obtain single-shot X-ray diffraction over a large momentum transfer $Q$ range and an ePix detector located at a larger distance is employed to resolve the speckle pattern's contrast with higher $Q$ resolution. A diode is used to measure the intensity of the direct beam on a single-shot basis. The droplets are injected with a droplet dispenser and cooled by evaporative cooling. In order to extract information about the dynamics, the exposure time is varied by changing the X-ray pulse duration $\delta t$ from 10 to 120 fs. **b** In the case where molecular motion is slower than the exposure time (top), the scattering pattern should exhibit high speckle contrast in reciprocal space. If the molecules move during the exposure time (bottom) the speckle contrast will be reduced, making the scattering pattern smoother

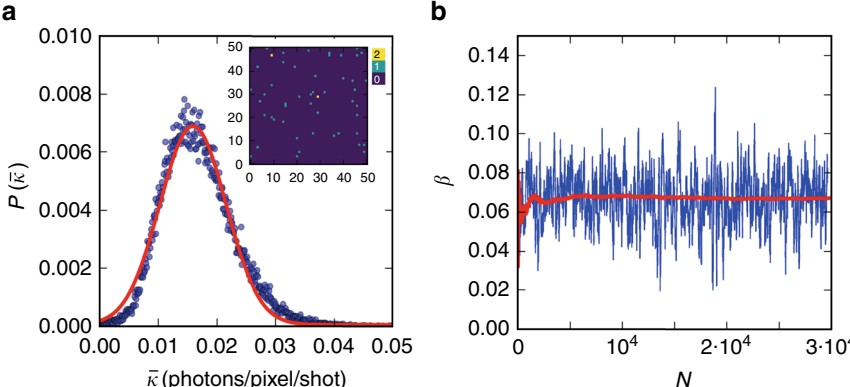

**Fig. 2** Speckle contrast analysis. **a** The mean photon density probability distribution of $3 \cdot 10^4$ shots recorded at $T = 296$ K with pulse duration $\delta t = 50$ fs. In the inset is shown a fraction of the ePix detector for a single shot with $\bar{k} = 1.5 \cdot 10^{-2}$ photons/pixel, which consists mainly of pixels with 1 photon count (green) and 2 photon counts (yellow). The solid red line depicts a Gaussian fit. **b** The speckle contrast $\beta$ as a function of number of shots $N$. Here is shown the running average over 120 shots (blue) and the cumulative average (red)

$Q$, allowing the distinction between frozen and supercooled liquid droplets on a shot-to-shot basis (see Methods section). The speckle patterns are recorded at the first diffraction peak, centred at $Q = 1.95$ Å$^{-1}$, using a second detector (ePix100)[33] located at a larger distance from the sample (see Methods section).

The X-ray diffraction pattern obtained from the CSPAD does not contain any dynamical information, since the speckle pattern cannot be resolved at this sample-to-detector distance and thereby reflects the ensemble average of the static structure factor of the system. On the other hand, the speckle pattern obtained from the ePIX detector is sensitive to the dynamic structure factor and the different pulse durations introduce the correlation time. By analysing the contrast of the speckle patterns, one can therefore extract information about the atomic motion occurring during the exposure time. A speckle pattern arises from interference between wavefronts that originate from the scattering of a coherent X-ray beam by the electron density of the molecules, which in the case of water is dominated by the oxygen atoms. Any change in their exact atomic positions will be reflected in the speckle pattern in reciprocal space and the speckle pattern thus reflects the instantaneous distribution of the positions of the molecules (Fig. 1b). Using sequential XPCS, one can follow the motion in real space by recording the changes in the speckle pattern[18]. Previous XPCS investigations of water measured diffusive dynamics during the high-to-low density transition, which in the ultra-viscous regime occurs on the order of seconds[22]. To probe dynamics in the sub-100 fs regime, we use the XSVS[23–25] approach in the ultrafast regime and vary the exposure time $\delta t$, which is performed by varying the FEL pulse duration instead of the detector exposure time. In this case, as $\delta t$ becomes comparable to or longer than the timescale of interest, the real-space arrangement of atoms is "blurred" and so is the corresponding speckle pattern (Fig. 1b). The underlying dynamics can thus be probed by evaluating the speckle contrast as a function of $\delta t$. In order to obtain statistics on the speckle contrast we probe different droplets and estimate the contrast on a single shot basis. By calculating the cumulative average of the speckle contrast we average over several different environments at a given correlation time $\delta t$.

**Speckle contrast analysis.** The single-shot speckle contrast can be obtained by analysing the scattering intensity distribution, which is expected to follow a Gamma distribution[34] whereas the shot-noise (quantum noise) due to the measurement follows a Poisson

distribution. In order to account for both effects and quantify the speckle contrast the negative binomial distribution (i.e., a convolution of the Gamma distribution with a Poisson distribution) is used

$$P(k, \bar{k}, M) = \frac{\Gamma(k + M)}{\Gamma(M)\Gamma(k + 1)} \left(1 + \frac{M}{\bar{k}}\right)^{-k} \left(1 + \frac{\bar{k}}{M}\right)^{-M} \quad (1)$$

where $k$ is the photon count of each pixel per shot, $\bar{k}$ is the average number of photons per pixel per shot and $M$ is the number of modes, which is related to the speckle contrast by $\beta = 1/M$.

The photon density distribution, $\bar{k}$, of $3 \cdot 10^4$ shots is shown in Fig. 2a, where the inset shows an example of a single shot measured with the ePix detector, which mainly contains one- and two-photon events. Due to the very low cross-section of water in the hard X-ray regime, we followed earlier work[34–39] and developed an analytical approach to estimate the contrast on a single-shot basis based on the negative binomial distribution. By solving Eq. (1) for $k = 1$ and $k = 2$ we get the expressions

$$P(1) \equiv P(k = 1, \bar{k}, M) = M \left(1 + \frac{M}{\bar{k}}\right)^{-1} \left(1 + \frac{\bar{k}}{M}\right)^{-M}$$

$$P(2) \equiv P(k = 2, \bar{k}, M) = \frac{M(M + 1)}{2} \left(1 + \frac{M}{\bar{k}}\right)^{-2} \left(1 + \frac{\bar{k}}{M}\right)^{-M}$$

Here the $P(1)$ and $P(2)$ is the number of 1 and 2 photon counts correspondingly, divided by the number of pixels. By defining the ratio $R_{12} \equiv \frac{P(2)}{P(1)}$ and solving this expression for $1/M$ we obtain the following analytical contrast estimator

$$\beta \equiv \frac{1}{M} = \frac{2 \cdot R_{12} - \bar{k}}{\bar{k}(1 - 2 \cdot R_{12})} \quad (2)$$

which relies solely on the number of one- and two-photon counts and allows estimating the speckle contrast at very low photon counts $\bar{k}$ on a single-shot basis. The running average of $\beta$ over 120 shots and the corresponding cumulative average over single shots is shown in Fig. 2b. The analytical estimator is compared to an alternative estimator, based on the maximum likelihood which utilizes the full photon histogram and not just $k = 1$ and $k = 2$, which yields identical results (see Supplementary Note 1).

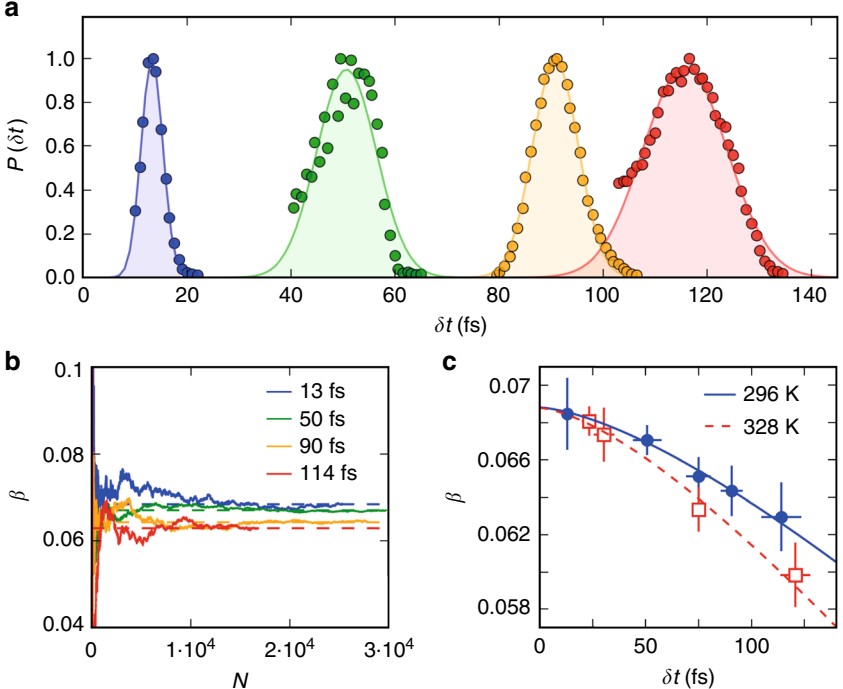

**Fig. 3** Pulse duration dependence. **a** The temporal probability distribution of X-ray shots for variable target pulse durations $\delta t$. **b** The cumulative average of the speckle contrast $\beta$ for increasing number of shots $N$ at $T = 296$ K for different target pulse durations as indicated in the legend. The dashed lines indicate the total average. **c** Speckle contrast $\beta$ as a function of pulse duration $\delta t$ for two different temperatures $T = 296$ K and 328 K. The maximum contrast $\beta_0$ for the current settings is estimated ($\beta_0 = 0.069 \pm 0.001$) by its value at $\delta t = 0$. The vertical error bars are the standard error and the horizontal the FWHM of the pulse duration distribution. The solid and dashed lines are fits to the experimental data

To control for dynamics induced by the X-ray pulse, the contrast $\beta$ was also analysed as a function of incident photon density (see Supplementary Notes 1 and 2). The deposited energy is absorbed on a sub-femtosecond timescale through the photo-electric effect, can generate photoelectrons which transfer energy to secondary electrons and through a cascade of processes influence the water molecules, leading to thermalization[40]. Even though the deposited energy density can lead to a temperature rise occurring on a timescale of several picoseconds to nanoseconds[41], we do not observe any heating effects within the pulse duration (see Supplementary Note 2). Therefore, with the current experiment the ground state of the sample before thermalization is still imprinted in the speckle pattern, thereby extending the diffraction-before-destruction approach[42] in probing dynamics. On the sub-100 fs timescale, the electronic excitation may involve ionization events and hot electron cascades, which however do not appear to contribute significantly due to the low fluence (see Supplementary Note 3). Since $\beta$ is found to be independent of the incident flux (see Supplementary Note 1), we conclude that on our experimental timescales (<150 fs) the measurements are within a non-perturbative regime. This observation is complemented by additional analysis of the $Q$ position of the first diffraction maximum as function of fluence; this peak is the most sensitive to the temperature (see Supplementary Note 2). Since the first diffraction maximum in $I(Q)$ appears independent of the fluence over nearly two orders of magnitude, we conclude that thermalization effects do not affect our measurements. Furthermore, cumulative radiation effects can be excluded, as every FEL shot interacts with a new water droplet. Therefore we conclude that we have no detectable influence by the incident beam on the water dynamics on the timescale probed here.

**Pulse duration and temperature dependence.** To extract the timescale of water motion in the sub-100 fs regime we varied the

FEL pulse duration $\delta t$, thereby varying the exposure time, from ~10 to 120 fs. The temporal probability distributions for different pulse durations are shown in Fig. 3a, where the data used are within the FWHM of each distribution, as discussed in detail in the see Supplementary Note 4 and 5. The cumulative average of the speckle contrast $\beta$ is shown in Fig. 3b for different pulse durations. We observe that the cumulative average of $\beta$ converges after ~$10^4$ FEL shots, with a signal-to-noise ratio that allows us to resolve the differences in $\beta$ as function of pulse duration, in agreement with analytical estimations (see Supplementary Note 6). These differences become more pronounced at higher temperatures, as can be seen in Fig. 3c, where the contrast as a function of the pulse duration $\delta t$ is shown. The contrast $\beta_0 = 0.069 \pm 0.001$ (i.e., the contrast at $\delta t = 0$ fs) is estimated at $T = 296$ K with a Gaussian fit (solid line) and is in agreement with the fit at $T = 328$ K (dashed line) and consistent with analytical estimations (see Supplementary Note 7). For both temperatures, $\beta$ decays for longer pulses within the standard error shown by the error bars. This is an indication of molecular motion within the pulse duration, which leads to a reduction of $\beta$ by ~8% at 296 K and ~12% at 328 K for the longest pulse duration of 115 and 120 fs, respectively.

In order to investigate the effect of temperature on molecular motion, we studied a broader temperature range from $T = 253$ K to 328 K for a fixed pulse duration $\delta t = 75$ fs. The temperature dependence of the angularly integrated scattering intensity is shown in Fig. 4a as function of momentum transfer $Q$. The position of the first diffraction maximum shifts to lower $Q$ upon cooling (Fig. 4b) which correlates with the increase of tetrahedral coordination[27,31,43]. Figure 4c shows the cumulative average of $\beta$ calculated on a single-shot basis for different temperatures (see Methods section). The corresponding average values for each temperature are shown in Fig. 4d, where the error bars are the standard error at each temperature. We observe that $\beta$ increases

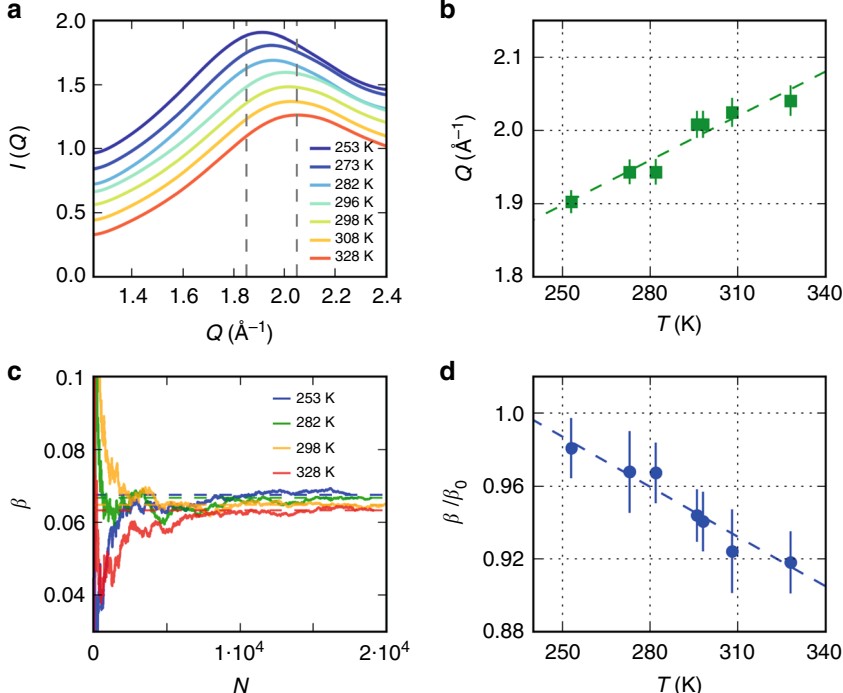

**Fig. 4** Temperature dependence. **a** Temperature dependence of the angularly integrated scattering intensity $I(Q)$ as a function of momentum transfer $Q$. The dashed lines indicate the $Q$-range over which the speckle contrast was investigated. **b** $Q$ value of the maximum of $I(Q)$, where the dashed line highlights the shift of the peak to lower $Q$ upon cooling. The errorbars is the standard error. **c** Cumulative average of the speckle contrast $\beta$ over number of X-ray laser shots $N$ for different temperatures as indicated in the figure. The dashed lines indicate the total average. **d** Corresponding average values of $\beta/\beta_0$, with error bars corresponding to the standard error. The dashed line highlights the increase of the speckle contrast upon cooling, which is an indication of gradual slowing down of the molecular motion within the exposure time $\delta t = 75$ fs

upon cooling, which indicates that the observed dynamics within 75 fs are becoming slower at lower temperatures, as detailed in the discussion section.

**Computational results**. To gain insight into the possible atomic-level mechanisms underlying our sub-100 fs experimental observations, we utilize molecular dynamics (MD) simulations using the TIP4P/2005[28] and MB-pol[29] water models. TIP4P/2005 shows very good agreement with real water for most properties, as exemplified by, e.g., the temperature of maximum density and the diffusion coefficient[28]. MB-pol is a flexible water model which includes polarizability and many-body effects through accurate fitting to high-level quantum chemical data for up to trimers[29,44].

In order to illustrate the magnitude of molecular displacements, snapshots from the TIP4P/2005 simulations are shown in Fig. 5a for the experimentally relevant timescales overlaid as a function of time. Within 120 fs the molecules are displaced from their initial positions, but have not yet reached their closest neighbours, a prerequisite for diffusive motion[45,46]. In order to compare the MD simulations with the experimental observations, i.e., the loss of speckle contrast $\beta$, the intermediate scattering function $F(Q, t)$ is computed

$$F(Q,t) = \frac{1}{N}\left\langle \sum_{i=1}^{N}\sum_{j=1}^{N} e^{iQ\cdot[r_i(t)-r_j(0)]} \right\rangle \quad (3)$$

where the angular brackets denote the ensemble average and $r_i(t)$ and $r_i(0)$ are the coordinates at time $t$ of an oxygen atom with index $i$ and the coordinates at time $t = 0$ of an oxygen atom $j$, respectively. The computed $F(Q, t)$ is normalised to $t = 0$ fs and averaged over the $Q$ range from 1.85 to 2.05 Å$^{-1}$, as shown in

Fig. 5b for temperatures ranging from 250 to 330 K. In agreement with previous MD studies, the fast sub-100 fs regime preludes the onset of the cage effect, when water molecules reach their nearest neighbours, after which diffusion on longer length- and time-scales sets in refs. [45,46].

## Discussion

The reduction of speckle contrast as function of the pulse duration (Fig. 3) is an indication of molecular motion on the timescale of the exposure time, probed at momentum transfer $Q = 1.95$ Å$^{-1}$. The two main peaks in the structure factor of water at 1.95 and 2.85 Å$^{-1}$ can be related to specific distances in the pair-correlation function[27,47,48]. Although there is interference in the Fourier components of the distances building the structure factor, a relationship with inverse momentum transfer to distance applies. The interference effect can be seen in the shift of the 1.95 Å$^{-1}$ peak with temperature, related to changes in the amplitude of the 4.5 Å correlation[27,43,49]. Since the 4.5 Å correlation is enhanced in tetrahedral structures, detecting the speckle contrast at $Q = 1.95$ Å$^{-1}$ provides sensitivity mainly toward dynamics involving tetrahedral structures.

Examining a broad range of temperatures (Fig. 4) we find that the observed motion becomes faster with increasing temperature, which is consistent with the simulation results of the intermediate scattering function $F(Q, t)$ (Fig. 5b). In order to compare the experiment with simulations, the speckle contrast $\beta$ is related to the intermediate scattering function $F(Q, t)$ by the Siegert relation, which in the case of XSVS yields[23–25]

$$\beta(Q, \delta t) = 2 \cdot \beta_0 \int_0^{\delta t} \left(1 - \frac{t}{\delta t}\right)|F(Q,t)|^2 \frac{\mathrm{d}t}{\delta t} \quad (4)$$

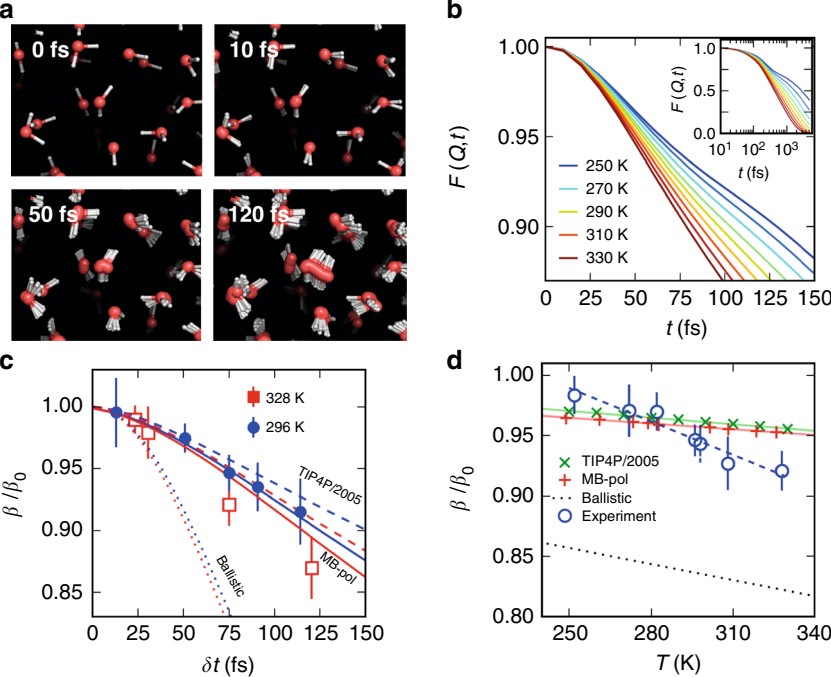

**Fig. 5** Molecular dynamics simulations and comparison to experiment. **a** Snapshots from the simulations for different exposure times comparable to the experiment. **b** The intermediate scattering function $F(Q,t)$ obtained from TIP4P/2005 at different temperatures. The inset depicts $F(Q,t)$ over longer times in logarithmic scale. **c** Comparison of the normalized speckle contrast between theory and experiment. The symbols are the experimental data at two different temperatures (circles 296 K, squares 328 K). The lines are simulated values using TIP4P/2005 (dashed) and MB-pol (solid), while the dotted lines correspond to the purely ballistic case at temperatures 300 K (blue) and 330 K (red). **d** Comparison of the speckle contrast as a function of temperature as estimated from the experiment (circles) with simulations (crosses MB-pol, x's TIP4P/2005) and the purely ballistic case (dotted line) at $\delta t$ = 75 fs. The lines depict linear fits to highlight the difference in slope between experiment and simulations. The errorbars in **c** and **d** correspond to the standard error

where $\delta t$ corresponds to the pulse duration and $\beta_0$ to the maximum contrast for a given experimental condition. Figure 5c shows the results obtained using Eq. (4) at temperatures $T = 300$ K (blue) and $T = 330$ K (red) with MB-pol (full lines) and TIP4P/2005 (dashed). The simulated $\beta$ with MB-pol agrees with the experiment within the error bars at 296 K whereas the TIP4P/2005 model exhibits a slower decay. At 328 K the experiment decays more rapidly as compared to the simulations; although again better agreement is observed for MB-pol. In addition, we show the case of pure ballistic motion, assuming a simple thermal model[50] (see Supplementary Note 8). In this case (dotted curves in Fig. 5c) $\beta$ decays significantly faster and features a significantly smaller variation with temperature. The experimental contrast variation surprisingly follows the ballistic motion only up to around 25 fs and then decays much more slowly. This is an indication that even though the sub-150 fs dynamics of water is often described as purely ballistic, the H-bonding influences the molecular motion by resisting distortions from the equilibrium position, which is often referred to as cage effects[45,51]. Since the probed $Q$-range makes the measurement specifically sensitive to the motion of tetrahedral structures involving strong H-bonds, the deviation from ballistic motion is amplified.

The temperature dependence of $\beta$ at fixed $\delta t = 75$ fs is compared to the simulated values in Fig. 5d. In the present case, both MD models describe very well the observed speckle contrast below $T = 290$ K, although the temperature-dependent change is more pronounced in the experiment. In addition, the ballistic model exhibits a significantly lower contrast than the experiment, indicating that the dynamics at 75 fs are already severely influenced by the cage effects, that come into play presumably due to

oxygen-oxygen oscillations and low-frequency intermolecular modes.

Although the trends are qualitatively well reproduced between the experimental and simulated data, there are important quantitative differences that we can utilize to test a hypothesis that could explain the data. Figure 6a shows a schematic of the different dynamical regimes probed by the intermediate scattering function. The early dynamics is related to the purely ballistic regime whereas the longer timescales reflect molecular diffusion. The current experiment probed the intermediate range, where the water molecules exhibiting ballistic motion are influenced by the neighbouring molecules via H-bonds, resulting in intermolecular modes that increase the occupancy time within the first solvation shell, referred to as cage effects. Specifically we see that the transition from the purely ballistic regime occurs already within 25 fs.

In the current hypothesis, we propose that the tetrahedral structures result in strong cage effects influencing the dynamics. Both the amount of tetrahedral structures contributing to the signal, as well as the dynamics related to the dissolution of the tetrahedral cage will affect the contrast variation. Here, we initially discuss the difference between the TIP4P/2005 simulations and experimental data. From static X-ray scattering measurements it is known that the amplitude of the 4.5 Å correlation is well represented by the TIP4P/2005 model over the current temperature range[27,52]. Could the difference observed here indicate that the cage dynamics of real water is faster at higher temperatures, as well as slower than the MD simulations at lower temperatures? Liquid water exhibits a tendency of forming tetrahedral regions which increase upon cooling, giving signatures

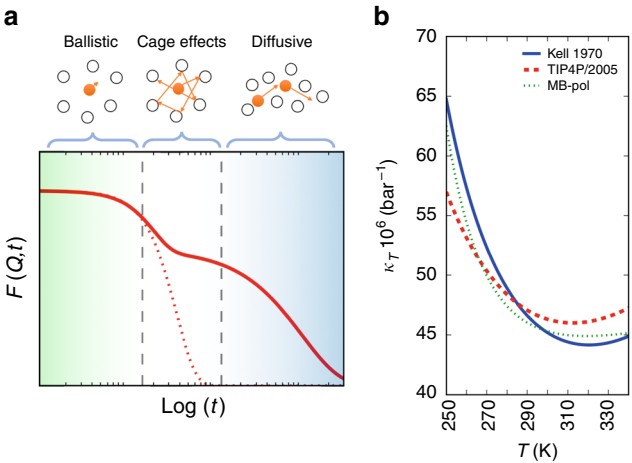

**Fig. 6** A schematic representation of the different regimes of water dynamics. **a** The intermediate scattering function $F(Q,t)$ is depicted as a function of the logarithm of time. The curves correspond to the experiment (solid), and the pure ballistic case (dotted). The vertical lines highlight the three different regimes: the early ballistic thermal-like motion, the cage effect regime and the diffusive regime. The current experiment probes the timescales between the ballistic regime and the cage effects. **b** The isothermal compressibility exhibits similar trends as the observed dynamics, when comparing experiments[60] with simulations[29,31]

in the long-range pair correlations[2,53,54] and in the low-$Q$ region in X-ray scattering[49,55,56]. Therefore one may expect that the cage dynamics become slower, as these regions grow and more time is required for rearrangements.

Using high-energy X-ray scattering with a large $Q$-range, the pair-correlation function can be derived very accurately[47,52]. It has been seen that the peak position corresponding to $r = 4.5$ Å correlation follows a normal linear expansion, similar to the first coordination shell upon increasing temperatures up to 310 K. However, at temperatures above 310 K the tetrahedral structures are no longer well-defined in terms of angles, contrary to the predictions of the TIP4P/2005 model, which also exhibits linear behaviour at higher temperatures[52]. This trend could imply that indeed the timescale for the cage effects in real water becomes much shorter than what is seen in the TIP4P/2005 model[2]. Toward the supercooled region, it has been shown that the low-$Q$ region in the X-ray scattering signal increases faster in water in comparison to the TIP4P/2005 model[57] implying a slower tetrahedral cage dynamics in water in comparison to the simulations. Based on these observations, we would expect the crossing-over in the cage dynamics between measured and TIP4P/2005 simulated water observed in Fig. 5d.

This hypothesis can be tested further with the temperature dependence of the isothermal compressibility ($\kappa_T$) which is a thermodynamic response function representing density or volume fluctuations[1]. The anomalous properties of water have been linked to the formation of extended tetrahedral structures[2,58], which, when appearing, take up more space than a more disordered arrangement. Fluctuating tetrahedral structures therefore lead to increased $\kappa_T$. This is the reason why $\kappa_T$ increases upon cooling below 319 K, where a minimum is located. One would expect that the fluctuations in and out of tetrahedral structures would be related to the tetrahedral cage dynamics. If a crossing between measured water and simulated water exists in the cage dynamics temperature dependence (Fig. 5d), we would expect that this is also the case for $\kappa_T$. Figure 6b shows the $\kappa_T$ temperature dependence obtained from experiments[59] and

simulations[28,29]. Indeed what is seen is that $\kappa_T$ for TIP4P/2005 compared to real water is higher when it is hot and lower when it is cold with a crossing around 280 K, which is consistent with the crossing in terms of dynamics (Fig. 5d). Furthermore, this observation is consistent with the MB-pol prediction being closer to the experimental value. This is further expected since the MB-pol model contains three-body interactions that will strengthen collective fluctuations of tetrahedral arrangements extending over several molecules. We expect similar effects from other models which also include three-body interactions, such as the E3B[60], which is furthermore parameterized for low-temperature phases of water[61]. These observations support the hypothesis that we observe the influence of tetrahedral cage dynamics in the speckle contrast measurements at $Q = 1.95$ Å$^{-1}$ and we foresee that these classes of experiments can be used to learn more about the time scale of the fluctuations extending to longer times.

The observed dynamics can be related to infrared spectroscopic signatures of liquid and supercooled water. Previous pump-probe investigations indicate that the OH-stretch frequency fluctuations of liquid water reflect an underdamped H-bond oscillation with 170 fs period[9]. This is consistent with the MD results, indicating that within the 120 fs pulse duration the water molecules do not yet exhibit diffusive behaviour but are affected by caging effects[45,46]. Therefore, the two techniques, XSVS and infrared spectroscopies, highlight complementary aspects of the water dynamics, by utilizing different experimental observables. XSVS results can also be compared to the dynamics obtained by 2D-IR spectroscopy at slightly supercooled temperatures[62], which reports on the loss of frequency correlation in the OH stretch vibration. Those investigations emphasize picosecond time-scale components that are attributed to H-bond network rearrangements[10–13] since limitations in temporal resolution in the IR regime make it difficult to precisely quantify the initial decay on the sub-150 fs timescale of the present study. An additional advantage of the current experiments using hard X-rays is that they permit probing directly the oxygen dynamics and do not rely on the correlation between structural changes and spectroscopic observables. Finally, the role of the collective H-bond dynamics has also been indicated by 2D Raman-THz investigations probing spectroscopically the THz modes in liquid water, which indicate the presence of H-bond heterogeneities occurring on a 100 fs timescale[17]. This observation is consistent with the picture we present here, where the observed sub-150 fs dynamics are not reproduced by a simple model with individual ballistic-like atomic motion, but instead arise from collective motion influenced by the H-bond network.

One can also measure the equilibrium dynamics using neutron spin-echo[6,7] and inelastic X-ray scattering[8]. In the case of neutron spin-echo the cross section is dominated by the hydrogens and therefore the measured dynamics relates mainly to hydrogen self-diffusion. Therefore, neutron spin-echo can complement the XSVS approach, which probes the oxygen motion. Inelastic X-ray scattering also probes the oxygen dynamics in the frequency domain, by resolving the incident photon energy. One of the advantages of this approach is that one can probe the longer time dynamics, although typically by using larger sample thickness. On the other hand, the XSVS time-domain implementation presented here can be applied to small droplets and probe the dynamics in the sub-100 fs using ultrashort X-ray pulses before thermalization occurs. In addition, by using femtosecond X-ray pulses one can also study liquid water in the deeply supercooled temperatures below the homogeneous nucleation temperature by outrunning crystallization using micron-sized or smaller droplets[27,49].

Summarizing, we have extracted the speckle contrast of liquid water as a function of pulse duration from 10–120 fs and over a temperature range from $T = 253$–328 K. As the pulse duration

increases, the speckle contrast decays due to molecular motion within the exposure time. The experimental values are compared with a simple model featuring thermal ballistic-like motion, which yields much more rapid decay of the speckle contrast. The discrepancy indicates that even though the sub-150 fs regime in water is often attributed to pure ballistic-like motion, it is actually influenced by the cage effects after ~25 fs. In addition, the dynamics are compared to those obtained from MD simulations of liquid water using the TIP4P/2005 force-field[28] and MB-pol[29]. We find that, even though the obtained speckle contrast decays near room temperature are consistent with the experiment, the MD simulations underestimate the dynamics at higher temperatures and exhibit a weaker temperature dependence. The better representation of the experimental data by MB-pol indicates the importance of many-body effects and polarizability in the sub-150 fs regime. Given the extraordinary sensitivity of the employed XSVS method to probe subtle changes on ultrafast timescales, the presented experimental data provide an unprecedented quantification of the equilibrium dynamics of liquid water in the sub-150 fs regime. It would be fascinating to extend the current study and capture the intermediate scattering function over longer timescales using a novel split-and-delay approach, which is currently being developed at several X-ray laser facilities world-wide, based on measuring the speckle contrast of a pair of ultra-short pulses with variable separation up to nanoseconds[63–67].

## Methods

**Experimental conditions.** The experimental data were recorded at the X-ray Correlation Spectroscopy (XCS) instrument of the Linac Coherent Light Source (LCLS) at SLAC National Accelerator Laboratory, during beamtime LM98. The photon energy used was $E = 8.2$ keV ($\lambda \approx 1.5$ Å), with a repetition rate of 120 Hz using Self-Amplified-Spontaneous-Emission (SASE) lasing. The pulse duration $\delta t$ for each FEL pulse was estimated with the X-Band Transverse Deflecting Cavity (XTCAV)[68] diagnostic on a single-shot basis. The beam was focused by a set of beryllium refractive lenses to a spot size of $s = 2$ μm diameter. A Si(111) monochromator was utilized to narrow the energy bandwidth and improve the beam longitudinal coherence quality ($\Delta E/E = 1.4 \cdot 10^{-4}$). The average pulse intensity on the sample after the Si(111) monochromator is estimated to vary between $10^9$ and $10^{10}$ photons/pulse depending on the pulse duration. The CSPAD detector was used with $194 \times 185$ pixels per unit, where four units were used in a standard $2 \times 2$ geometry[32] and placed at a short distance ($d = 150$ mm). The ePix100[33] detector was used to resolve the speckle pattern and features $352 \times 384$ pixels per unit with pixel size $50 \times 50$ μm². Two detector units were used at a larger sample-to-detector distance ($d = 1.3$ m) covering the momentum transfer range $Q = 1.85-2.05$ Å⁻¹. This large distance is necessary to resolve the speckle pattern, the characteristic speckle size[33] of which can be estimated by $w = \frac{\lambda d}{s}$, where $s$ is the focus size, $\lambda$ the wavelength and $d$ the sample-detector distance. For the given experimental settings $w = 98$ μm which is ~2 times larger than the pixel size and fulfils the required criterion.

**Sample environment.** The sample environment used to measure the water jet has been used previously to study water under deeply supercooled conditions[27,31]. Here, we used two different settings: one for supercooled conditions ($T = 280-250$ K) and one for higher temperatures ($T = 290-330$ K). For supercooling, a series of droplets with 95 μm diameter were generated by a droplet dispenser in the vacuum chamber. Pressures on the order of $10^{-2}$ mbar were reached by using three turbo-molecular pumps with pumping speed 300 L s⁻¹ each, while the jet was directed toward a liquid nitrogen cold trap. The droplet temperature was varied by adjusting the distance between the nozzle and the X-ray interaction region, thereby effectively changing the cooling duration. The frequency of the piezoelectric actuator that was used to drive the jet breakup into droplets was varied between 21 and 22 kHz. A microscope was used to estimate the droplet diameter in situ (95 μm), the droplet spacing (285 μm), and the velocity (~12 m s⁻¹). The droplet temperature was estimated as described elsewhere[27,31] using the Knudsen theory of evaporation, which has in addition been confirmed by MD simulations[69]. For reaching higher temperatures, the jet was used in a continuous mode (not droplet mode) with a dispenser with diameter 100 μm and was used under Helium environment at atmospheric pressure. The dispenser was heated locally by a resistor and the temperature was measured in the interaction region with a thermocouple.

**Data analysis.** One of the most critical aspects of the analysis was selecting the X-ray laser shots for which certain experimental criteria were fulfilled: the X-rays hit the centre of the droplet and the recorded X-ray diffraction indicated that the droplet is in the liquid state and not frozen, the X-ray pulses were within the desired pulse duration, the pulse intensity was sufficient to estimate the contrast with precision and the photon energy was within a certain range in order to maintain the desired $Q$ resolution. The details of imposing those constraints and the contrast metric estimators are detailed in the Supplementary Notes 1 and 2.

**Simulations.** Molecular dynamics simulations were performed using the TIP4P/2005[28] and MB-pol[29] water models. TIP4P/2005 is a four-site water model consisting of a Lennard-Jones site for the oxygen atom, and three charge sites, and features a general parametrization for simulating the entire phase diagram of water. First, equilibration runs were performed in the NVT ensemble for 2 ns, at equilibrium densities determined previously for atmospheric pressure for temperatures between 250 and 340 K in steps of 10 K. The Nosé-Hoover thermostat was used together with a 2 fs simulation time step. All simulation boxes consisted of 45,000 molecules, corresponding to the approximate box side of 10 nm. Additional NVT simulations lasting 2 ns were subsequently performed and snapshots containing atomic positions and velocities were stored in double precision every 200 ps. These snapshots were finally used as starting points for ten statistically independent 10 ps long NVE trajectories at each temperature, where the time step was 1 fs and double-precision was used for accurate energy conservation. The Gromacs 5.0.4 simulation package was used for all trajectories. Classical molecular dynamics simulations of water interacting via the MB-pol potential[29] were also performed. MB-pol is a polarizable potential which explicitly includes short-range two-body and three-body terms fitted to high-level quantum-mechanical interactions. Cubic simulation boxes consisting of 1510 water molecules were equilibrated in the NPT ensemble at 1 bar pressure for 100 ps with an integration step of 0.5 fs. Additional 40 ps long simulations with 0.2 fs integration step were run to reach equilibrium densities. Finally, 40 ps long trajectories were collected in the NVE ensemble at each temperature to study the dynamics. These simulations were performed using the MB-pol interface to the i-PI wrapper[70] available at http://paesanigroup.ucsd.edu/software/mbpol_ipi.html.

**Data availability.** The authors declare that the data supporting the findings of this study are available within the paper and its Supplementary Information files or from the corresponding author on reasonable request.

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

## Acknowledgements

We acknowledge financial support from the European Research Council (ERC advanced grant WATER under project no. 667205) and the Swedish Research Council (VR) through grants 2013-8823 and 2016-04875. F.P. was additionally supported by the Swiss National Science Foundation (fellowship P2ZHP2 148666). F.L. and G.G. thank The Hamburg Centre for Ultrafast Imaging for support. K.T.W. was supported by the Icelandic Research Fund. Use of the LCLS, SLAC National Accelerator Laboratory, is supported by the U.S. Department of Energy, Office of Science, Office of Basic Energy Sciences under Contract No. DE-AC02-76SF00515. All simulations were performed on resources provided by the Swedish National Infrastructure for Computing (SNIC) at the PDC Centre for High Performance Computing (PDC-HPC) and High Performance Computer Center North (HPC2N). We would like to thank Elias Diesen for useful discussions and comments on the manuscript.

## Author contributions

F.P., G.G., and A.N. conceived and designed the experiments. F.P., T.J.L., J.A.S., F.L., S.N., P.H., and M.S. performed the data analysis. G.C., T.J.L., K.T.W., L.G.M.P., and F.P.

performed and analysed the molecular dynamics simulations and modelling. Sa.S., T.S., M.S., W.R., R.A.-M., Y.F., D.Z., and A.R. operated the XCS instrument. A.S., H.P., F.P., K. H.K., K.A.-W., T.M., H.O., D.N., and J.K. designed, built, and operated the sample environment. Si.S., A.E., and D.N. performed the e-logging during the experiments. F.P. and A.N. wrote the manuscript with input from all authors.

## Additional information

**Competing interests:** The authors declare no competing interests.

