## [Peer Review File · Nature Communications]

Reviewer #1 (Remarks to the Author):

Report

The authors present the results of a speckle visibility experiment of liquid water performed at an x-ray free-electron laser. Analyzing the loss in speckle contrast as a function of pulse duration the authors claim to observe equilibrium dynamics of the ballistic motion of water molecules on sub-100 fs timescales.

The experiment is technically very challenging. Demonstrating the feasibility of XSVS on atomic/molecular length- and fs time-scales constitutes a real breakthrough for the technique. It opens the door for a whole class of experiments which seek to study fluctuations in (photo) excited matter. The authors definitely deserve credit for this wonderful achievement.

However, in the current form and with the current twist the paper is scientifically not sound and I cannot recommend it for publication. The main claim of the authors is that the measured loss in speckle contrast represents equilibrium dynamics of water. They continue by saying that with this, XSVS is superior to e.g. THz pump-probe experiments which perturb the system.

It is a simple exercise to estimate the amount of energy deposited inside the sample for a single shot of $1e10$ photons of energy 8.2 keV in a water volume of $2x2x95$ microns. The numbers are large and it becomes evident that the sample is in a highly perturbed state. I would guess that this energy corresponds to increases in temperatures of 100s to 1000 K (of course on ps-ns time scales). On time scales of 100fs the perturbation is electronic in nature and the implications for the dynamics inside the H-network are completely unclear. The disagreement shown in Fig 4 may also simply be connected to the perturbed/ionized surrounding created during the X-ray pulse duration. The observed temperature dependence of the decay in contrast indicates that the ground state of the sample is still imprinted in the first 100 fs when the sample is subject to hot electron cascades ionizing the water network before it starts to explode/vaporize. I guess this is the really interesting observation made here.

The authors base their claim of observing equilibrium dynamics onto the absence of fluence dependence of the contrast shown in Fig. 2c in the supplement. However, Fig. 1c supplement suggests that the variation in incidence intensity used by the authors is not large. So, intensities in the low-damage regime (i.e. with effective dT below 10 K) cannot be studied in such experiments. The correlation between photon intensity and pulse duration remains also unclear.

In conclusion, very nice experiment and very nice data. However, suggesting that such an intense X-ray pulse leaves the sample in an equilibrium state is questionable. Fig. 2 c is not a convincing argument. Moreover, aiming for equilibrium dynamics other techniques such as neutron spin-echo or inelastic X-ray scattering may also yield similar information.

As a last remark: I suggest to change the title to something more scientific.

Reviewer #2 (Remarks to the Author):

Review of Perakis et al. "Probing the dynamics of water on the fly"

The authors present their results measured at the LCLS XFEL using a novel technique called X-ray Speckle Visibility Spectroscopy, where the coherent speckle pattern from a liquid water sample at various temperatures is measured for different X-ray pulse durations. The results of these measurements provide information on the velocity of the water molecules, which they authors compare to the results of MD simulations, resulting in several discrepancies.

First I would like to congratulate the authors on their work. This is a very nice experimental and technical effort that uses a novel technique to probe the dynamics of water. I believe this work should be published in Nature Communications after resolution of some minor issues.

I have two issues I would like addressed before publication:

First I believe there is a lack of clarity for the reader with respect to the connection between the measurement and the analysis of the experimental data. The procedure to analyse the experimental measurement is clarified in the Supporting Information, but I believe some of this should be included in the Results section. For example equation 1 describes a negative binomial distribution, but the primary criterion for data evaluation is the speckle contrast. The speckle contrast is a function of the number of modes (M), but there is no explanation of what a mode is until the SI (pixels/speckle ?). I think a short, simple description of speckle contrast is missing. One simple addition which might clarify things could be an actual recorded 2D speckle pattern from the ePix ?

My second issue involves the discrepancy between the MD simulations and the experiment. In several places in the text there is mention that the experiment and theory match well for cooler temperatures (296 K) but diverge for warmer temperatures (328 K). While this might be statistically true within the error bars, it doesn't seem to describe the deviation between the trends of the theory and experiment. In Figure 4c the MD simulation always over-estimates the $F(Q,t)$ in comparison to the experiment for both temperatures. And in 4d the MD results cross the experimental measurement of the velocity at lower temperatures, but it's clear the over all trend of the two results have markedly different temperature dependences. From this I would be tempted to say agreement between experiment and theory isn't great. As a reader I would like to know what this means, with respect to the reliability of the MD model, and perhaps how the model fails in this regards ? Would a QM/MM model be more appropriate ? It also looks to me like the divergence between MD and experiment in 4c appears to get worse the longer the timescale, which is the opposite of what I might expect ?

Reviewer #3 (Remarks to the Author):

This paper describes the experimental and theoretical study of the dynamics of liquid water on very short (~100 fsec.) timescales. The authors use an elegant technique, employing the duration of x-ray pulses as “shutter time” to monitor the motion of water molecules, through the oxygen-oxygen diffraction intensity distribution. The speckle pattern, measured as a function of x-ray pulse duration, reflects how, at different temperatures, oxygen atoms move relative to each other in liquid water.

I was somewhat disappointed towards the end of the manuscript; rather than having learned something new about the dynamics of water, the authors have introduced a new technique in the study of water, and highlight substantial discrepancies between the experimental measurements and theory. It is not clear, however, whether those discrepancies originate from assumptions in the analysis of the experimental data, shortcomings of the theoretical model, or challenges in connecting the output of the simulations to the experimental observations.

More detailed comments:

In the introduction, the authors write: “For many systems the observed dynamics are close to the true ground state equilibrium, whereas in the case of liquid water the excitation of the OH stretch mode can result in coherent oxygen oscillations.” Likewise, towards the end of the manuscript, it is stated: “On the other hand, such oscillations can be induced by a pump-pulse in a pump-probe experiment, leading to coherent quantum beats.” These are very strange statements, which I do not understand: the coherence between the different water molecules is indeed induced by a pump

pulse in a pump probe experiment, but the appearance of underdamped oscillations for individual O-H groups is a property of the system, and is not "induced by a pump-pulse". It is made visible by a pump-pulse, and one could argue that perhaps the dynamics of water with one vibrationally excited O-H group is somehow different from water in the vibrational ground state, but the claim that the pump pulse induces motion in the water that are otherwise not present is nonsense - it synchronizes that motion, which would otherwise be asynchronous, to make it 'visible'. Such statements are unnecessary - the two techniques, time resolved x-ray and infrared spectroscopies, highlight different aspects of the same systems, and are therefore simply complementary.

First mentioned on page 3, please define CSPAD.

On page 3, caption of figure 1: the last sentence contains a redundancy: "...time (bottom) the speckle contrast will be reduced, making the scattering pattern smoother with a reduced speckle contrast."

On page 4, in the description of the experiment and the analysis, the main text needs a brief, clear description of how beta is extracted from the diffraction data. I have read the SI in sufficient detail to find a typo (" In this case, the average is 0.069 and the standard error is 0.02" where the error should read 0.002), but still was not able to understand exactly how the data is treated to obtain the values for beta. In all immodesty, I don't think my incapability of understanding what the authors did reflects on my mental abilities, but rather on the extremely concise way the data treatment is presented. This is a fundamental and critical point, as a clear description of data treatment is a prerequisite for the possibility of other researchers to reproduce, and verify, the results. A more detailed description of how beta-parameters are extracted from single shot, two-dimensional camera images has to be provided, at least in the SI.

On page 5, figure 2A: Why are there no data points on the short end of the distributions? The pulse length distribution seems to have a tail to short times. Why is this not plotted, and what would be the implication for the analysis? Are the data points collected on the low ends not used? This should be made very explicit; I appreciate that the authors list the four criteria used for filtering the data in the SI, but again: it should be made clear also in the main text, with at least a sentence or two, why the data looks the way it does in figure 2A.

On page 5, figure 2C: It is not clear why the Beta_0 is constrained to be the same for different temperatures; it is apparent that independent linear fits to the 296 and 328K data would provide different values for Beta/Beta_0 in the limit of zero pulse duration.

On page 6, figure 3: Just to satisfy my curiosity – not necessarily to be discussed in the manuscript: how about the temperature of maximum density at 277 K - would one not expect to see an effect of the maximum density?

On page 7, figure 4: The marked disagreement between experiment and theory, given the claim by the authors that XSVS should provide relatively direct access to molecular dynamics, would indicate that the level of theory is insufficient to describe adequately the short time dynamics of water. I do not understand, especially given the short timescales involved, why the authors didn't employ a higher level of theory. This would seem very straightforward, and would allow for much more meaningful statements about the technique, the theory and the behavior of the water molecules in short timescales.

February 7, 2017

Reviewer #1:

“The experiment is technically very challenging. Demonstrating the feasibility of XSVS on atomic/molecular length- and fs time-scales constitutes a real breakthrough for the technique. It opens the door for a whole class of experiments which seek to study fluctuations in (photo) excited matter. The authors definitely deserve credit for this wonderful achievement.”

We appreciate the positive feedback from the reviewer also his/her insight and suggestions for improving the current work. We address the reviewer’s comments as follows:

1. **“It is a simple exercise to estimate the amount of energy deposited inside the sample for a single shot of $1e10$ photons of energy 8.2 keV in a water volume of $2x2x95$ microns. The numbers are large and it becomes evident that the sample is in a highly perturbed state. I would guess that this energy corresponds to increases in temperatures of 100s to 1000 K (of course on ps-ns time scales). On time scales of 100fs the perturbation is electronic in nature and the implications for the dynamics inside the H-network are completely unclear. The disagreement shown in Fig 4 may also simply be connected to the perturbed/ionized surrounding created during the X-ray pulse duration. The observed temperature dependence of the decay in contrast indicates that the ground state of the sample is still imprinted in the first 100 fs when the sample is subject to hot electron cascades ionizing the water network before it starts to explode/vaporize. I guess this is the really interesting observation. (...) The authors base their claim of observing equilibrium dynamics onto the absence of fluence dependence of the contrast shown in Fig. 2c in the supplement. However, Fig. 1c supplement suggests that the variation in incidence intensity used by the authors is not large.”**

This is a very useful and valid comment concerning the temperature rise occurring at slower timescales (ps to ns), although the electronic excitation should not change the structure of the water network on the 100fs timescale, as the reviewer comments. In order to clarify this point in the manuscript we have introduced a new in-depth analysis of the intensity maximum Q position as a function of fluence for the for the longest pulses ($\delta t = 120$ fs), as obtained from the CSPAD detector. In this case, the data were analysed over a larger range spanning over an nearly two orders of magnitude and no significant fluence dependence was observed. We included the following discussion:

“The deposited energy is absorbed on a sub-femtosecond timescale through the photoelectric effect, generating photoelectrons which transfer energy to secondary electrons and through a cascade of

CHEMICAL PHYSICS DIVISION

Anders Nilsson
Department of Physics
Stockholms universitet

AlbaNova University Center
Roslagstullsbacken 21
S-10691 Stockholm

Phone: +46 (0)8-553 786 37
E-mail: andersn@fysik.su.se

processes influence the water molecules, leading to thermalization⁴⁰. In our experiment the deposited energy density is on the order of 1 MJ/kg, which can lead to a temperature rise occurring on a timescale of several picoseconds to nanoseconds⁴¹. On the sub-100 fs timescale however, the electronic excitation may involve ionization events and hot electron cascades, but would not yet impact the structure and dynamics of the water molecules. Therefore, with the current experiment the ground state of the sample before thermalization is still imprinted in the diffraction pattern, thereby extending the diffraction-before-destruction approach^{42,43} in probing dynamics. Since β is found to be independent of the incident flux (see SI), we conclude that on our experimental timescales (<150 fs) the measurements are within a non-perturbative regime. This observation is complemented by additional analysis of the Q position of the first diffraction maximum as function of fluence; this peak is the most sensitive to the temperature (see SI). Since the first diffraction maximum in $I(Q)$ appears independent of the fluence over nearly two orders of magnitude, we conclude that thermalization effects do not affect our measurements.”

And in the Supplementary information:

“As discussed in the main text in order to estimate any possible beam induced heating effects, that can arise within the pulse duration we have analysed the Q -position of the first diffraction peak as a function of photon density. This analysis complements the analysis discussed at Fig. S2a, where the contrast is also analysed as a function of the photon density. In Fig. S2a we concluded that we do not observed any beam induced changes in the estimated contrast, as the contrast is constant for a broad range of photon densities, that corresponds to different pulse intensities. Here in addition, we perform as similar analysis for the longer pulse duration that was used $\delta t = 120$ fs and for two different temperatures, $T = 296$ K and $T = 328$ K, shown in Fig. S6. The y-range chosen matches Fig. 4b of the main manuscript and the x axis is the photon counts of the ePIX detector (in logarithmic scale), as shown in Fig. S2c, which is proportional to the incident energy. If there was a temperature rise occurring within the pulse duration, one would expect the Q to shift at higher values with increasing fluence, whereas here the Q position of the first diffraction maximum in $I(Q)$ appears independent of the fluence over an of magnitude.”

Figure S6 | The Q position of the first diffraction maximum of the angularly integrated intensity $I(Q)$ as a function of photon density \bar{k} for $\delta t = 120$ fs.

2. So, intensities in the low-damage regime (i.e. with effective dT below 10 K) cannot be studied in such experiments.”

In order to extend the current experimental approach to longer timescales, such as picoseconds and nanoseconds, one would need to utilize lower incident energy, as the reviewer suggests. One can estimate the conditions under which such regimes can be studied. We have introduced the following discussion in the SI addressing the reviewers comment:

“The reason that the data in Fig.S2 become dispersed in the low photon density range is because the limited number of shots available in this range, as can be seen from histogram in Fig.2a. By attenuating the beam one can also obtain with precision the speckle contrast at lower fluence, although a larger number of shots is needed.”

And a new section with the title *“signal-to-noise calculation”*:

“The required number of required shots at different photon densities can be estimated analytically by calculating the signal-to-noise ratio as a function of number of shots⁷, by the following expression:

$$\frac{\sigma_{\beta}}{\beta} = \frac{1}{\bar{k} \cdot \beta} \sqrt{\frac{2(1 + \beta)}{n_{pix} \cdot N}}$$

where n_{pix} is the number of pixels and N is the number of shots. To illustrate the scaling of the signal-to-noise ratio with number of shots we estimate the $\frac{\sigma_{\beta}}{\beta}$ in two different regime: for $\bar{k} = 10^{-2}$

photons/pixel/pulse, relating to the current settings, and for $\bar{k} = 10^{-3}$ photons/pixel/pulse, which could be related the estimated photon density using a split-and-delay that can be used to measure longer timescales.

Figure S3 | The estimated signal-to-noise ratio as a function of number of shots N for two conditions with photon density \bar{k} (photons/pixel/pulse).

Here, we use the following values of the current experiment $\beta = 0.07$ and $n_{\text{pix}} = 352 \times 384 \times 2$. One can see that, as presented in the main manuscript, for $\bar{k} \approx 10^{-2}$ one can reach a signal-to-noise ratio of 35 (which for $\beta = 0.07$ corresponds to $\sigma_{\beta} = 0.002$) after approximately 10^4 shots. For lower photon densities $\bar{k} \approx 10^{-3}$ approximately 10^6 shots are required per condition, which is still within the experimental capabilities of the current FEL sources. One can also improve this scaling by implementing a larger number of detectors covering a larger fraction of the azimuthal angle, thereby increasing n_{pix} , which will also be possible with the rapid advancements in the detector technology. Finally, by utilizing the higher repetition rate of the upcoming FEL sources one can dramatically improve the signal-to-noise ratio by measuring several orders of magnitude more shots at each experimental condition.”

3. “The correlation between photon intensity and pulse duration remains also unclear.”

We have introduced the following section in the SI:

“The photon density at each pulse duration depends on the LCLS performance at the given pulse duration. Here, we report the mean photon density for different pulse duration for $T = 296\text{K}$, which is obtained by a Gaussian fit of the distribution in each case.

Figure S7 | The mean photon density $\langle \bar{k} \rangle$ as a function of pulse duration δt for $T = 296\text{ K}$.

4. “Moreover, aiming for equilibrium dynamics other techniques such as neutron spin-echo or inelastic X-ray scattering may also yield similar information.”

We have introduced references to neutron diffraction and inelastic X-ray scattering and extend the discussion about the comparison of the different approaches at a separate section with the title “Connections to other techniques”.

“One can also measure the equilibrium dynamics using neutron spin-echo^{6,7} and inelastic x-ray scattering⁸. In the case of neutron spin-echo the cross section is dominated by the hydrogens and therefore the measured dynamics relates to the hydrogen self-diffusion. Therefore, neutron spin-echo can

complement the XSVS approach, which probes the oxygen motion. Inelastic x-ray scattering also probes mainly the oxygen dynamics in the frequency domain, by resolving the incident photon energy. One of the advantages of this approach is that one can probe the longer time dynamics, although typically by using larger sample thickness. On the other hand, the XSVS time-domain implementation presented here can be applied to small droplets and probe the dynamics in the sub-100 fs using ultrashort x-ray pulses before thermalization occurs. In addition, by using femtosecond x-ray pulses one can also study liquid water in the deeply supercooled temperatures below the homogeneous nucleation temperature by outrunning crystallization using micron-sized or smaller droplets^{28,50}.”

5. **“I suggest to change the title to something more scientific.”**

We have updated the title to emphasize the new understanding on water dynamics, as follows:

“Coherent x-rays reveal the influence of cage effects on ultrafast water dynamics”

Reviewer #2:

“First I would like to congratulate the authors on their work. This is a very nice experimental and technical effort that uses a novel technique to probe the dynamics of water. I believe this work should be published in Nature Communications after resolution of some minor issues.”

We are glad that the reviewer appreciates the difficulties and challenges of the current experimental study and address her/his comments in the following section.

1. **“The procedure to analyse the experimental measurement is clarified in the Supporting Information, but I believe some of this should be included in the Results section... I think a short, simple description of speckle contrast is missing. One simple addition which might clarify things could be an actual recorded 2D speckle pattern from the ePix ?”**

We have introduced a figure in the main manuscript, accompanied with a new section that explains in more detail the extraction of the speckle contrast from the 2D image:

“The photon density distribution, \bar{k} , of $3 \cdot 10^4$ shots is shown in Fig. 2a, where the inset shows an example of a single shot measured with the ePix detector, which mainly contains one- and two-photon events. Due to the very low cross-section of water in the hard x-ray regime, we followed earlier work³⁶⁻³⁹ and developed an analytical approach to estimate the contrast on a single-shot basis based on the negative binomial distribution. By solving equation (1) for $k = 1$ and $k = 2$ we get the expressions:

$$P(1) \equiv P(k = 1, \bar{k}, M) = M \left(1 + \frac{M}{\bar{k}}\right)^{-1} \left(1 + \frac{\bar{k}}{M}\right)^{-M}$$

$$P(2) \equiv P(k = 2, \bar{k}, M) = \frac{M(M+1)}{2} \left(1 + \frac{M}{\bar{k}}\right)^{-2} \left(1 + \frac{\bar{k}}{M}\right)^{-M}$$

Here the $P(1)$ and $P(2)$ is the number of 1 and 2 photon counts correspondingly, divided by the number of pixels. By defining the ratio $R_{12} \equiv \frac{P(2)}{P(1)}$ and solving this expression for $1/M$ we obtain the following analytical contrast estimator

$$\beta \equiv \frac{1}{M} = \frac{2 \cdot R_{12} - \bar{k}}{\bar{k}(1 - 2 \cdot R_{12})}, \quad (2)$$

which relies solely on the number of one- and two-photon counts and allows estimating the speckle contrast at very low photon counts \bar{k} on a single-shot basis. The running average of β over 120 shots and the corresponding cumulative average over single shots is shown in Fig. 2b. The analytical estimator is compared to an alternative estimator, based on the maximum likelihood which utilizes the full photon histogram and not just $k = 1$ and $k = 2$, which yields identical results (see SI)."

Figure 2 | Speckle contrast analysis. **a**, The mean photon density probability distribution of $3 \cdot 10^4$ shots recorded at $T = 296$ K with pulse duration $dt = 50$ fs. In the inset is shown a fraction of the ePix detector for a single shot with $\bar{k} = 1.5 \cdot 10^{-2}$ photons/pixel, which consists mainly of pixels with 1 photon counts (green) and 2 photon counts (yellow). **b**, The speckle contrast β as a function of number of shots N . Here is shown the running average over 120 shots (blue) and the cumulative average (red).

2. “... Discrepancy between the MD simulations and the experiment... As a reader I would like to know what this means, with respect to the reliability of the MD model, and perhaps how the model fails in this regards? Would a QM/MM model be more appropriate? It also looks to me like the divergence between MD and experiment in 4c appears to get worse the longer the timescale, which is the opposite of what I might expect?”

We have address the discrepancy between the MD simulations by including additional simulations using MB/pol, as well as simplifying the analysis by comparing the speckle contrast of the experiment with the simulations (see similar comment from reviewer #3). Finally, we discuss that even though MD might not reproduce the experiment in the sub-150fs regime, where the water dynamics are influenced by cage effects due to the hydrogen bond network, it can potentially follow more closely the diffusive regime as suggested by infrared spectroscopic investigations. Thereby, this discrepancy between MD and experiment is presumably limited in the transition between pure ballistic and diffusive regimes.

Reviewer #3:

“I was somewhat disappointed towards the end of the manuscript; rather than having learned something new about the dynamics of water, the authors have introduced a new technique in the study of water, and highlight substantial discrepancies between the experimental measurements and theory.”

We understand the input from the reviewer and have made some major changes to highlight the new understanding about water dynamics originating from the current work. We address the reviewer’s comments point-by-point in the following section:

1. “..rather than having learned something new about the dynamics of water”

By changing the title, as reviewer #1 suggested, we emphasize the influence of cage effects on the fast sub-100fs regime of liquid water. This regime is often assigned to pure ballistic like-motion, but as our measurement indicates the dynamics is already influenced by the hydrogen bond network after 25 fs. In the discussion section, we have also introduced a new figure and describe the different regimes and propose a hypothesis that described our observations:

“Although the trends are qualitatively well reproduced between the experimental and simulated data, there are important quantitative differences that we can utilize to test a hypothesis that could explain the data. Fig. 6a shows a schematic of the different dynamical regimes probed by the intermediate scattering function. The early dynamics is related to the purely ballistic regime whereas the longer timescales reflect molecular diffusion. The current experiment probed the intermediate range, where the water molecules exhibiting ballistic motion are influenced by the neighbouring molecules via H-bonds, resulting in intermolecular modes that increase the occupancy time within the first solvation shell, referred to as cage effects. Specifically we see that the transition from the purely ballistic regime occurs already within 25 fs.

Figure 6 | A schematic representation of the different regimes of water dynamics. (a) The intermediate scattering function $F(Q, t)$ is depicted as a function of the logarithm of time. The curves correspond to the experiment (solid), and the pure ballistic case (dotted). The vertical lines highlight the three different regimes: the early ballistic thermal-like motion, the cage effect regime and the diffusive regime. The current experiment probes the timescales between the ballistic regime and the cage effects. (b) The isothermal compressibility exhibits similar trends as the observed dynamics, when comparing experiments⁵³ with simulations^{29,30}.

In the current hypothesis, we propose that the tetrahedral structures result in strong cage effects influencing the dynamics. Both the amount of tetrahedral structures contributing to the signal, as well as the dynamics related to the dissolution of the tetrahedral cage will affect the contrast variation. Here, we initially discuss the difference between the TIP4P/2005 simulations and experimental data. From static x-ray scattering measurements it is known that the amplitude of the 4.5 Å correlation is well represented by the TIP4P/2005 model over the current temperature range^{28,54}. Could the difference observed here indicate that the cage dynamics of real water is faster at higher temperatures, as well as slower than the MD simulations at lower temperatures? Liquid water exhibits a tendency of forming

tetrahedral regions which increase upon cooling, giving signatures in the long-range pair correlations^{2,55,56} and in the low- Q region in x-ray scattering^{50,57,58}. Therefore one may expect that the cage dynamics become slower, as these regions grow and more time is required for rearrangements.”

2. **“It is not clear, however, whether those discrepancies originate from assumptions in the analysis of the experimental data, shortcomings of the theoretical model, or challenges in connecting the output of the simulations to the experimental observations.”... “I do not understand, especially given the short timescales involved, why the authors didn't employ a higher level of theory. This would seem very straightforward, and would allow for much more meaningful statements about the technique, the theory and the behavior of the water molecules in short timescales.”**

In order to address the reviewers comments we have simplified the analysis and compare now the simulations directly with the experimental observable, the speckle contrast. In addition, we have employed a higher level of theory, by performing simulations using MB/pol water model which is flexible, includes polarizability and short-range 3-body interactions. Here is detailed the updated text and figure:

“Fig. 5c shows the results obtained using equation (4) at temperatures $T = 300$ K (blue) and $T = 330$ K (red) with MB-pol (full lines) and TIP4P/2005 (dashed). The simulated β with MB-pol agrees with the experiment within the error bars at 296 K whereas the TIP4P/2005 model exhibits a slower decay. At 328 K the experiment decays more rapidly as compared to the simulations ; although again better agreement is observed for MB-pol. In addition, we show the case of pure ballistic motion, assuming a simple thermal model⁵¹ (see SI). In this case (dotted curves in Fig. 5c) β decays significantly faster and features a significantly smaller variation with temperature. The experimental contrast variation surprisingly follows the ballistic motion only up to around 25 fs and then decays much more slowly. This is an indication that even though the sub-150 fs dynamics of water is often described as purely ballistic, the H-bonding influences the molecular motion by resisting distortions from the equilibrium position, which is often referred to as cage effects^{46,52}. Since the probed Q -range makes the measurement specifically sensitive to the motion of tetrahedral structures involving strong H-bonds, the deviation from ballistic motion is amplified.

The temperature dependence of β at fixed $\delta t = 75$ fs is compared to the simulated values in Fig. 5d. In the present case, both MD models describe very well the observed speckle contrast below $T = 290$ K, although the temperature-dependent change is more pronounced in the experiment. In addition, the ballistic model exhibits a significantly lower contrast than the experiment, indicating that the dynamics at 75 fs are already severely influenced by the cage effects, that come into play presumably due to oxygen-oxygen oscillations and low-frequency intermolecular modes.

Figure 5 | Molecular dynamics simulations and comparison to experiment. **a**, Snapshots from the simulations for different exposure times comparable to the experiment. **b**, The intermediate scattering function $F(Q,t)$ obtained from TIP4P/2005 at different temperatures. The inset depicts $F(Q,t)$ over longer times in logarithmic scale. **c**, Comparison of the normalized speckle contrast between theory and experiment. The symbols are the experimental data at two different temperatures (circles 296 K, squares 328 K). The lines are simulated values using TIP4P/2005 (dashed) and MB-pol (solid), while the dotted lines correspond to the purely ballistic case at temperatures 300 K (blue) and 330 K (red). **d**, Comparison of the speckle contrast as a function of temperature as estimated from the experiment (circles) with simulations (crosses MB-pol, x's TIP4P/2005) and the purely ballistic case (dotted line) at $\delta t = 75$ fs. The lines depict linear fits to highlight the difference in slope between experiment and simulations.

(...) Using high-energy x-ray scattering with a large Q -range, the pair-correlation function can be derived very accurately^{48,54}. It has been seen that the peak position corresponding to $r = 4.5$ Å correlation follows a normal linear expansion, similar to the first coordination shell upon increasing temperatures up to 310 K. However, at temperatures above 310 K the tetrahedral structures are no longer well-defined in terms of angles, contrary to the predictions of the TIP4P/2005 model, which also exhibits linear behavior at higher temperatures⁵⁴. This trend could imply that indeed the timescale for the cage effects in real water becomes much shorter than what is seen in the TIP4P/2005 model². Towards the supercooled region, it has been shown that the low- Q region in the x-ray scattering signal increases faster in water in comparison to the TIP4P/2005 model⁵⁹ implying a slower tetrahedral cage dynamics in water in comparison to the simulations. Based on these observations, we would expect the crossing-over in the cage dynamics between measured and TIP4P/2005 simulated water observed in Fig. 5d.”

3. “..the coherence between the different water molecules is indeed induced by a pump pulse in a pump probe experiment, but the appearance of underdamped oscillations for individual O-H groups is a property of the system, and is not "induced by a pump-pulse". It is made visible by a pump-pulse, and one could argue that perhaps the dynamics of water with one vibrationally excited O-H group is somehow different from water in the vibrational ground state, but the claim that the pump pulse induces motion in the water that are otherwise not present is nonsense - it synchronizes that motion, which would otherwise be asynchronous, to make it ‘visible’. Such statements are unnecessary - the two techniques, time resolved x-ray and infrared spectroscopies, highlight different aspects of the same systems, and are therefore simply complementary.”

We have modified this discussion point, in accordance to the reviewer’s suggestion. The referenced parts now read:

“H-bond breaking and forming dynamics have been proposed to occur on a picosecond timescale. Furthermore, the intermolecular dynamics of water molecules can be probed in the THz regime, where the low-frequency modes in the range 50-300 cm^{-1} are attributed to H-bond oscillations¹⁴. The sensitivity of these THz modes to the local H-bond network has been seen both from simulations¹⁵ and

experiments^{16,17}. One of the challenges when using spectroscopic techniques, both in the IR- and THz-regime, is relating the spectroscopic observable to a specific lengthscale and motion in the liquid. The temporal resolution of most pump-probe implementations is limited due to the longer wavelengths in the IR and THz-regime as compared to x-rays, which makes it difficult to probe the sub-100 fs regime where initial molecular displacements occur.”

and the final discussion:

“The observed dynamics can be related to infrared spectroscopic signatures of liquid and supercooled water. Previous pump-probe investigations indicate that the OH-stretch frequency fluctuations of liquid water reflect an underdamped H-bond oscillation with 170 fs period⁹. This is consistent with the MD results, indicating that within the 120 fs pulse duration the water molecules do not yet exhibit diffusive behaviour but are affected by caging effects^{46,47}. Therefore, the two techniques, XSVS and infrared spectroscopies, highlight complementary aspects of the water dynamics, by utilizing different experimental observables.”

4. “define CSPAD”

We have introduced the following definition: “Cornell-SLAC Pixel Array Detector”

5. On page 3, caption of figure 1: the last sentence contains a redundancy: “...time (bottom) the speckle contrast will be reduced, making the scattering pattern smoother with a reduced speckle contrast.”

Thanks for point this out, we have removed the redundancy.

6. “..find a typo (“ In this case, the average is 0.069 and the standard error is 0.02” where the error should read 0.002)”

Thanks, we have corrected it.

7. “A more detailed description of how beta-parameters are extracted from single shot, two-dimensional camera images has to be provided, at least in the SI”

We have introduced a new section in the main manuscript elaborating on the details of how the beta-parameters are extracted from a single shot. See similar comment from reviewer #2.

8. “figure 2A: Why are there no data points on the short end of the distributions? The pulse length distribution seems to have a tail to short times. Why is this not plotted, and what would be the implication for the analysis? Are the data points collected on the low ends not used? This should be made very explicit; I appreciate that the authors list the four criteria used for filtering the data in the SI, but again: it should be made clear also in the main text, with at least a sentence or two, why the data looks the way it does in figure 2A.”

As the reviewer correctly points out this is related to the four criteria used for filtering at the SI. We have introduced the following discussion in the main text:

“The temporal probability distributions for different pulse durations are shown in Fig. 3a., where the data used are within the FWHM of each distribution, as discussed in detail in the SI.”

9. “It is not clear why the Beta_0 is constrained to be the same for different temperatures; it is apparent that independent linear fits to the 296 and 328K data would provide different values for Beta/Beta_0 in the limit of zero pulse duration.

We have updated the figure and independently extracted beta_0 for the two temperatures and the corresponding errors. On the y-axis we show instead the beta values directly and a Gaussian form has been used to capture in more precision the β_0 .

“The $\beta_0 = 0.069 \pm 0.001$ (i.e. the contrast at $\delta t = 0$ fs) is estimated at $T = 296$ K with a Gaussian fit and is in agreement with those at $T = 328$ K and consistent with analytical estimations (see SI). “

10. “how about the temperature of maximum density at 277 K - would one not expect to see an effect of the maximum density?”

This is a very interesting suggestion. We believe that the compressibility is more appropriate in terms of volume and density fluctuations related to the dynamics. We have included the following discussion:

“This hypothesis can be tested further with the temperature dependence of the isothermal compressibility (κ_T) which is a thermodynamic response function representing density or volume fluctuations¹. The anomalous properties of water have been linked to the formation of extended tetrahedral structures^{2,60}, which, when appearing, take up more space than a more disordered arrangement. Fluctuating tetrahedral structures therefore lead to increased κ_T . This is the reason why κ_T increases upon cooling below 319 K, where a minimum is located. One would expect that the fluctuations in and out of tetrahedral structures would be related to the tetrahedral cage dynamics. If a crossing between measured water and simulated water exists in the cage dynamics temperature dependence (Fig. 5d), we would expect that this is also the case for κ_T . Fig. 6b shows the κ_T temperature dependence obtained from experiments³³ and simulations^{29,30}. Indeed what is seen is that κ_T for TIP4P/2005 compared to real water is higher when it is hot and lower when it is cold with a crossing around 280 K, which is consistent with the crossing in terms of dynamics (Fig. 5d). Furthermore, this observation is consistent with the MB-pol prediction being closer to the experimental value. This is further expected since the MB-pol model contains three-body interactions that will strengthen collective fluctuations of tetrahedral arrangements extending over several molecules. We expect similar effects from other models which also include three-body interactions, such as the E3B3⁶¹ which is furthermore parameterized for low-temperature phases of water⁶². These observations support the hypothesis that we observe the influence of tetrahedral cage dynamics in the speckle contrast measurements at $Q = 1.95 \text{ \AA}^{-1}$ and we foresee that these classes of experiments can be used to learn more about the time scale of the fluctuations extending to longer times.”

Sincerely yours,
Anders Nilsson

Reviewer #1 (Remarks to the Author):

I am still struggling with the main message of the paper. As said before it remains unclear, also in the revised version, to which extend this represents an equilibrium situation. The revised version now contains some of my wording from the first report – which is of course not the idea of a revised version. The authors now claim:

On the sub-100 fs timescale however, the electronic excitation may involve ionization events and hot electron cascades, but would not yet impact the structure and dynamics of the water molecules.

without giving further evidence for this statement. However, it is also clear that the answer to this question can only be addressed by a deeper understanding of the ultrafast X-ray matter interaction mechanisms – which is beyond the scope of this paper.

So, overall, the revised version is much clearer and accessible now and at the end this paper will spark the necessary discussion in the scientific community about the opportunities and limitations of these kind of experiments. As such it represents an important contribution and will definitely draw attention in the community. I do not agree with some of the statements of the authors but I highly recommend publication of this work.

One last comment (also to the Editor): the mathematics around Eq. (2) seems to have been published in arxiv arXiv:1710.01015 already by others – even with the same notation. So a reference may do the charm.

Reviewer #2 (Remarks to the Author):

I would like to thank the authors for addressing my comments. I believe the changes they have made in response to the reviewers' comments has improved and clarified the manuscript and I recommend this for publication.

Reviewer #3 (Remarks to the Author):

I very much appreciate the efforts of the authors to address the concerns of me and the other reviewers.

I am afraid, though, going over the text and the comments of reviewer 1 in particular, two related concerns have arisen:

“In our experiment the deposited energy density is on the order of 1 MJ/kg, which can lead to a temperature rise occurring on a timescale of several picoseconds to nanoseconds. On the sub-100 fs timescale however, the electronic excitation may involve ionization events and hot electron cascades, but would not yet impact the structure and dynamics of the water molecules.”

I am not worried about heating - I think the authors exclude that convincingly, also by demonstrating the fluence independence of the experiments. I am much more worried about the local ionization events and hot electron cascades, which the authors confirm are present. The problem is that the claim that the x-ray pulse is too short to do more than take a “snapshot” of the water is not consistent, or rather: in direct contradiction, with the claim that the transients are determined by “cage effects on ultrafast water dynamics”, i.e. the motion of water molecules. If there are ionization events - and these will occur on attosecond time scales - these will for sure affect the interaction potential between water molecules, and thereby modify the motion of those same water molecules. This is not a global effect, therefore temperature is not of concern: it is a highly localized effect, but it is important because the same water molecules that are being probed by the x-rays are being affected by those same x-rays. This is also consistent with the absence of fluence dependence: the signal will not become fluence dependent until these ionization effects start to affect one another; this will not happen until very high excitation densities are reached. Therefore the absence of fluence dependence is no proof of a non-perturbative experiment.

This brings me to a second problem, closely related to the first, namely that of ergodicity. Under thermodynamic equilibrium, the ensemble average response of the system over space should be indistinguishable from that recorded over time. With the short X-ray pulse the authors measure an ensemble average (the beam profile being much larger than the molecular diameter) and with long X-ray pulse they average also over time. As such, it's not clear to me that one would expect the change in the diffraction pattern when changing the x-ray pulse duration. The question is then also when that change “saturates”, and what is the typical timescale associated with that saturation mean? The fact that a difference is observed between these two (time-averaged and spatially averaged) measurements means that ensemble and time average are different. This seems to contradict ergodicity. Again, this seems to imply that the authors are measuring a non-equilibrium situation in their experiment.

The label “d” is missing in figure 5.

Reviewer #1:

1. **“The revised version now contains some of my wording from the first report – which is of course not the idea of a revised version. The authors now claim: *“On the sub-100 fs timescale however, the electronic excitation may involve ionization events and hot electron cascades, but would not yet impact the structure and dynamics of the water molecules”*. without giving further evidence for this statement. However, it is also clear that the answer to this question can only be addressed by a deeper understanding of the ultrafast X-ray matter interaction mechanisms – which is beyond the scope of this paper.”**

We appreciate the comment from the reviewer and agree that directly probing and modeling the ionization events and electron cascades is beyond the scope of this paper. In order to qualitatively address the comment of both reviewers 1 and 3, we estimate the probability of ionization for the given experimental conditions as a function of fluence. This calculation is included as a new section in the supplementary information:

“Ionization probability

In order to estimate the magnitude of possible contributions due to ionization events that can occur during the pulse duration, we estimate the ionization probability as a function of fluence [13].

The number of molecules N_{mol} in the probed volume can be estimated by taking into account the number density of water ($n = 33.3679 \cdot 10^{27} \text{ m}^{-3}$) as well as the focus size (radius $r = 1 \text{ }\mu\text{m}$) and sample thickness ($w = 98 \text{ }\mu\text{m}$). This yields:

$$N_{mol} = n \cdot V = n \cdot (w \cdot \pi r^2) \approx 1.03 \cdot 10^{13} \text{ molecules}$$

CHEMICAL PHYSICS DIVISION

Anders Nilsson
Department of Physics
Stockholms universitet

AlbaNova University Center
Roslagstullsbacken 21
S-10691 Stockholm

Phone: +46 (0)8-553 786 37
E-mail: andersn@fysik.su.se

The total number of absorption events is proportional to the number of photons per unit area I_0 and the sample thickness w , where the proportionality constant is the absorption coefficient μ ($\mu \approx 1 \text{ mm}^{-1}$ at 8.2 keV for water): This is also the number of photoelectrons emitted:

$$N_e = I_0 \cdot w \cdot \mu$$

The I_0 is defined by the ratio of incident energy ($\approx 10 \mu\text{J}$) over the area πr^2 , which is in the order of 10^9 photons/ μm^2 . The ionization probability per μm^2 is defined as the number of photoelectrons over the number of molecules in the probed volume:

$$P_{\text{ionization}} = \frac{N_e}{N_{\text{mol}}}$$

The ionization probability as a function of fluence is shown in Fig.S8.

Figure S8 | The probability of ionization as a function of fluence. For the current experimental conditions the fluence is estimated between 10^9 to 10^{10} photons/pulse resulting 10^{-5} to 10^{-4} ionization events per molecule per μm^2 .

For fluences of the order of 10^9 to 10^{10} photons/pulse the ionization probability is between 10^{-5} to 10^{-4} electrons per molecule, assuming single photon absorption. In other words this corresponds to one ionization event per 10^4 to 10^5 molecules, which is well beyond the sensitivity of the current experiment. At higher fluences however, which can be reached for example by utilizing the full (i.e. pink) beam of LCLS combined with a nanofocus, the ionization probability can become significant[13]. In addition, previous experimental [14] and theoretical [15] investigations indicate that each photoelectron will lead to many secondary electrons depending on the photon energy. However, since the ionization probability depends on the fluence (Fig.S8), any potential contributions to the dynamics would also involve fluence dependence. Since the speckle contrast which reflects the dynamics (Fig.S2) appears fluence-independent, we tentatively conclude that we do not observe any major contributions due to ionization.”

We have also modified the main manuscript which now reads:

“Even though the deposited energy density can lead to a temperature rise occurring on a timescale of several picoseconds to nanoseconds⁴², we do not observe any heating effects within the pulse duration (see SI). Therefore, with the current experiment the ground state of the sample before thermalization is still imprinted in the speckle pattern, thereby extending the diffraction-before-destruction approach⁴³ in probing dynamics. On the sub-100 fs timescale, the electronic excitation may involve ionization events and hot electron cascades, which however do not appear to contribute significantly due to the low fluence (see SI).”

2. **“One last comment (also to the Editor): the mathematics around Eq. (2) seems to have been published in arxiv arXiv:1710.01015 already by others – even with the same notation. So a reference may do the charm.”**

We have included the citation

Reviewer #3:

1. **I am not worried about heating - I think the authors exclude that convincingly, also by demonstrating the fluence independence of the experiments. I am much more worried about the local ionization events and hot electron cascades, which the authors confirm are present. The problem is that the claim that the x-ray pulse is too short to do more than take a “snapshot” of the water is not consistent, or rather: in direct contradiction, with the claim that the transients are determined by “cage effects on ultrafast water dynamics”, i.e. the motion of water molecules. If there are ionization events - and these will occur on attosecond time scales - these will for sure affect the interaction potential between water molecules, and thereby modify the motion of those same water molecules. This is not a global effect, therefore temperature is not of concern: it is a highly localized effect, but it is important because the same water molecules that are being probed by the x-rays are being affected by those same x-rays. This also consistent with the absence of fluence dependence: the signal will not become fluence dependent until these ionization effects start to affect one another; this will not happen until very high excitation densities are reached. Therefore the absence of fluence dependence is no proof of a non-perturbative experiment.**

We have addressed the comment, along with the corresponding comment from reviewer #1. As discussed in the new section in the supplementary information “***Ionization probability***” we estimate that with the current fluence we expect one ionization event per 10^4 to 10^5 molecules. In addition, we do not only probe the fluence dependence of the structure (*Fig.S8*) but of the dynamics too, by analysing the fluence dependence of the contrast at the longest pulse duration (*Fig.S2*), which appears fluence-independent. Since the number of ionization events depends on fluence (*Fig.S8*), the impact of such events would show up as fluence dependence of the dynamics.

2. **This brings me to a second problem, closely related to the first, namely that of ergodicity. Under thermodynamic equilibrium, the ensemble average response of the system over space should be indistinguishable from that recorded over time. With the short X-ray pulse the authors measure an ensemble average (the beam profile being much larger than the molecular**

diameter) and with long X-ray pulse they average also over time. As such, it's not clear to me that one would expect the change in the diffraction pattern when changing the x-ray pulse duration. The question is then also when that change "saturates", and what is the typical timescale associated with that saturation mean? The fact that a difference is observed between these two (time-averaged and spatially averaged) measurements means that ensemble and time average are different. This seems to contradict ergodicity. Again, this seems to imply that the authors are measuring a non-equilibrium situation in their experiment.

The reviewer here suggests that the experiment may result in different time and spatial averages, contradicting ergodicity. This might be a misunderstanding, since with the XSVS approach used here we have not reached "saturation" neither in space nor in time, but instead we probe the speckle pattern for different correlation times. The ensemble average of the speckle contrast in this case is performed by probing over 10^4 different droplets (per condition) on a single shot basis, as described in the main manuscript. We have introduced the following discussion in the main manuscript to further clarify this point (p.3, line 9):

"The x-ray diffraction pattern obtained from the CSPAD does not contain any dynamical information, since the speckle pattern cannot be resolved at this sample-to-detector distance and thereby reflects the ensemble average of the static structure factor of the system. On the other hand, the speckle pattern obtained from the ePIX detector is sensitive to the dynamic structure factor and the different pulse durations introduce the correlation time."

just before the existing section:

"By analysing the contrast of the speckle patterns, one can therefore extract information about the atomic motion occurring during the exposure time. A speckle pattern arises from interference between wavefronts that originate from the scattering of a coherent x-ray beam by the electron density of the molecules, which in the case of water is dominated by the oxygen atoms. Any change in their exact atomic positions will be reflected in the speckle pattern in reciprocal space and the speckle pattern thus reflects the instantaneous distribution of the positions of the molecules (Fig. 1b). Using sequential XPCS, one can follow the motion in real space by recording the changes in the speckle pattern¹⁹. Previous XPCS investigations of water measured diffusive dynamics during the high-to-low density transition, which in the ultra-viscous regime occurs on the order of seconds²³. To probe dynamics in the sub-100fs regime, we use the XSVS²⁴⁻²⁶ approach in the ultrafast regime and vary the exposure time δt , which is performed by varying the FEL pulse duration instead of the detector exposure time. In this case, as δt becomes comparable to or longer than the timescale of interest, the real-space arrangement of atoms is "blurred" and so is the corresponding speckle pattern (Fig. 1b). The underlying dynamics can thus be probed by evaluating the speckle contrast as a function of δt ."

which is then complemented by (p.4, line 10):

"In order to obtain statistics on the speckle contrast we probe different droplets and estimate the contrast on a single shot basis. By calculating the cumulative average of the speckle contrast we average over several different environments at a given correlation time δt ."

3. The label "d" is missing in figure 5

Thanks, we have included the label in the new version.

Reviewer #3 (Remarks to the Author):

I thank the authors for their efforts that have alleviated most of my concerns, and while some loose ends remain in this story, this should not prohibit publication of this very nice work.